# Supramolecular Protein-Polyelectrolyte Assembly at Near Physiological Conditions—Water Proton NMR, ITC, and DLS Study

**DOI:** 10.3390/molecules27217424

**Published:** 2022-11-01

**Authors:** Alexander Marin, Marc B. Taraban, Vanshika Patel, Y. Bruce Yu, Alexander K. Andrianov

**Affiliations:** 1Institute for Bioscience and Biotechnology Research, University of Maryland, Rockville, MD 20850, USA; 2Department of Pharmaceutical Sciences, School of Pharmacy, University of Maryland, Baltimore, MD 21201, USA

**Keywords:** protein–polyelectrolyte interactions, polyphosphazenes, supramolecular assembly, isothermal titration calorimetry, water proton transverse relaxation rate, dynamic light scattering, immunoadjuvant, vaccine delivery vehicle, model antigen

## Abstract

The in vivo potency of polyphosphazene immunoadjuvants is inherently linked to the ability of these ionic macromolecules to assemble with antigenic proteins in aqueous solutions and form physiologically stable supramolecular complexes. Therefore, in-depth knowledge of interactions in this biologically relevant system is a prerequisite for a better understanding of mechanism of immunoadjuvant activity. Present study explores a self-assembly of polyphosphazene immunoadjuvant—PCPP and a model antigen—lysozyme in a physiologically relevant environment—saline solution and neutral pH. Three analytical techniques were employed to characterize reaction thermodynamics, water-solute structural organization, and supramolecular dimensions: isothermal titration calorimetry (ITC), water proton nuclear magnetic resonance (*w*NMR), and dynamic light scattering (DLS). The formation of lysozyme–PCPP complexes at near physiological conditions was detected by all methods and the avidity was modulated by a physical state and dimensions of the assemblies. Thermodynamic analysis revealed the dissociation constant in micromolar range and the dominance of enthalpy factor in interactions, which is in line with previously suggested model of protein charge anisotropy and small persistence length of the polymer favoring the formation of high affinity complexes. The paper reports advantageous use of *w*NMR method for studying protein-polymer interactions, especially for low protein-load complexes.

## 1. Introduction

Supramolecular assembly of proteins and synthetic polyelectrolytes in aqueous solutions are gaining an increasing popularity in life sciences realm. Interactions of ionic polymers with proteins find innovative applications in diverse technological settings, such as protein purification and stabilization, delivery of biotech drugs, tissue engineering, preparation of bioactive multilayer coatings, and biosensing, and have been a subject of a number of recent reviews [1,2,3,4,5,6,7,8,9]. Perhaps one of the most captivating examples of the importance of such interactions in biomedicine is the mechanism of in vivo immunostimulation, which is enabled by an ionic polyphosphazene via spontaneous supramolecular assembly with vaccine antigens [10,11,12,13,14,15].

Poly[di(carboxylatophenoxy)phosphazene], PCPP is a biodegradable high molecular weight polyanion, which has demonstrated its immunoadjuvant potency in multiple animal models and in several clinical trials [10,11,16,17,18]. Structurally, it can be characterized by a high linear charge density resulting from two carboxylic acid side groups per repeat unit and small persistent length, which stems out from highly flexible phosphorus-nitrogen backbone [19]. The underlying fundamentals of PCPP adjuvanted vaccines include a spontaneous assembly of this polymer with antigenic proteins, which has been proven for numerous formulations [12,13,14,20,21,22,23,24,25]. Strong durability of the immune response to PCPP-containing vaccines also suggests the persistence of its complexes with antigens under physiological conditions, which inherently comprise salts in concentrations that have been considered detrimental for many non-covalent interactions [26,27,28]. However, studies on protein–polyelectrolyte interactions in physiological environment remain scarce and further physico-chemical and biophysical research is warranted to advance knowledge on complex formation under those conditions in general and gain better understanding of the mechanism of action for PCPP in particular. 

Present paper investigates supramolecular assembly of PCPP immunoadjuvant and well-characterized protein–hen egg lysozyme at near physiological conditions–phosphate-buffered saline (PBS), pH 7.4. The choice of lysozyme in the study has been driven by its frequent use as a model antigen in immunogenicity studies [29,30,31] and proven in vivo performance of PCPP when formulated with this protein [32]. Additionally, opposite charges of lysozyme (positive) and PCPP (negative) at pH 7.4 facilitated the electrostatic driven supramolecular assembly. A combination of three independent analytical techniques have been employed to study PCPP-lysozyme binding. Isothermal titration calorimetry (ITC) has been one the most important choices in studying protein-polyelectrolyte interactions and, more generally, to biomolecular recognition reactions due to repeatability, sensitivity, as well as for providing an important thermodynamic information [3,33,34]. Water proton NMR (*w*NMR) is a powerful tool, which monitors the signal of water protons and its transverse relaxation rate *R*_2_(^1^H_2_O) mediated by interactions with dissolved matter, was selected due to its non-invasive approach and proven sensitivity in the detection macromolecular aggregates and cluster of nanoparticulates [35,36,37,38,39,40]. Lastly, the results of the above analytical methods were cross-referenced with dynamic light scattering (DLS) data in an attempt to understand a potential link of the detected characteristics with physical state and dimensions of lysozyme-PCPP complexes. 

Here, we report the formation of supramolecular complexes of lysozyme and PCPP under the conditions mimicking physiological environment, and independent confirmation and characterization of the self-assembly process independently by ITC, *w*NMR, and DLS. The obtained results define these interactions as driven primarily by an enthalpic factor. This is in line with previously suggested ‘anisotropy’ model, which assumes binding of flexible polyelectrolyte chains with oppositely charge patches on the protein surface. Finally, for the first time, we demonstrate the advantageous utility of *w*NMR method for exploring protein-polyelectrolyte interactions.

## 2. Results

Interactions between lysozyme and PCPP at near physiological conditions (PBS, pH 7.4) were studied using three independent analytical techniques—ITC, *w*NMR, and DLS with the objective of comparing and cross-validating the results.

### 2.1. Isothermal Titration Calorimetry (ITC) Studies

ITC method has been employed as a convenient approach to evaluating thermodynamic parameters and avidity of binding between proteins and polyelectrolytes, although interactions, which take place under physiological pH and salt concentration, remain scarcely explored [3,33,41,42,43]. The titration of PCPP with lysozyme was carried out at near physiological conditions—PBS, pH 7.4. Figure 1A shows representative raw data for the titration, whereas Figure 1B,C compare titrations performed at two different concentrations—“Low” (0.125 and 0.5 mg/mL polymer and protein, correspondingly} and “High” (0.5 and 10 mg/mL). The results for both experiments aligned well when presented as a heat released per a single PCPP chain (Figure 1C). Both curves indicate a rapid decrease in the heat released when the molar ratio exceeded approximately 100 lysozyme molecules per PCPP chain.

Evaluation of ITC data using Single Set of Identical Sites (SSIS) approach revealed both negative enthalpy and entropy changes, dissociation constant of 5 × 10^−8^ M, as well as complex stoichiometry (n) at 165 protein-to-polymer molar ratio (Appendix A, Figure A1).

Reverse titration of lysozyme with PCPP under the same conditions showed abrupt release of heat at the beginning of the titration with no heat detected upon further addition of PCPP (Appendix A, Figure A2).

### 2.2. wNMR Studies

Self-assembly of lysozyme and PCPP was also explored using noninvasive analytical technique—water proton NMR (*w*NMR). Unlike conventional analytical NMR approaches, *w*NMR monitors the signal of H_2_O protons, and its transverse relaxation rate *R*_2_(^1^H_2_O) mediated by water-solute interactions could serve as a probe of solute physico-chemical status/structural organization. Due to overwhelmingly higher water concentration compared to lysozyme and PCPP, *R*_2_(^1^H_2_O) could be easily measured using low-field benchtop time-domain NMR instruments. To ensure that the observed changes indeed characterize the formation of complexes, *R*_2_(^1^H_2_O) dependence on the concentration of protein and polymer alone were first studied (Figure 2A,B, dotted and dashed lines, correspondingly). As seen from Figures, the results demonstrated the absence or very shallow effects of individual components on water proton transverse relaxation rates in the explored concentration range. In contrast, titration of PCPP with lysozyme showed a rapid increase in *R*_2_(^1^H_2_O) in the same concentration range (Figure 2A solid blue line). Contrary to this, changes observed when reverse titration was conducted were minimal (Figure 2B solid brown line). Of note, under the reverse titration conditions PCPP is added at rather high concentration (10 mg/mL), and quite probably is already clustered, thus, the increase in lysozyme to PCPP ratio has almost no effect on the *R*_2_(^1^H_2_O) value (Figure 2B,C, solid brown line). Figure 2C shows the results plotted as a function of molar protein-to-polymer ratios. The study was also performed when both components were mixed at equal volumes. In this case, the changes in water proton transverse relaxation rate were similar to those observed for the titration of polymer with the protein (Figure 2D).

### 2.3. Comparison of ITC, wNMR, and DLS Results—Potential Cooperativity Effect

The results of ITC and *w*NMR studies were cross-referenced with DLS characterization of self-assembly in lysozyme-PCPP system. The dependence of z-average hydrodynamic diameter on the protein-to-polymer molar ratios, as determined by DLS, shows two main regions for complexes (Figure 3A). At a lower lysozyme-to-polymer ratios the assemblies are characterized by nano-submicron scale dimensions, with diameters of complexes remaining practically unchanged (around 400 nm) when they contain 15–90 protein molecules per polymer chain (noted as region I in Figure 3A). The situation changes dramatically when more than 100 lysozyme copies are assembled on PCPP resulting in the formation of micron-size aggregates (region II, Figure 3A). Review of ITC data in the context of the above DLS results reveals that the threshold of 100 protein molecules per polymer chain also corresponds to the beginning of a dramatic drop in the reaction rate (region II in Figure 1B and Figure 3B). Furthermore, *w*NMR data also shows leveling off in *R*_2_(^1^H_2_O) values approximately at the same complex composition (Figure 3C). These results suggest that the formation of the aggregates at high lysozyme loadings leads to a dramatic reduction in the binding rate in the system. 

Comparison of *w*NMR and DLS data also reveals that the former method is able to detect changes in the range of protein-to-polymer molar ratios (up to 5), in which DLS shows unchanged dimensions similar to those of PCPP—around 60 nm (Figure 4). This indicates the formation of single chain complexes with insignificant size change, which can carry up to five protein molecules before the complex starts aggregating into multi-chain formations.

## 3. Discussion

The compendium of ITC, *w*NMR, and DLS data characterizes interactions between lysozyme and PCPP as a formation of submicron (60–400 nm) and micron sized (1 µm and higher) complexes with multiple protein molecules assembled onto a single polymer chain (Figure 5). All methods suggest that the transition to micron size aggregates take place when the excess of bound protein exceeds one hundred protein copies per polymer chain (Figure 3A,B). *w*NMR data also suggest that in the submicron size range the complex is first formed as a single polymer chain assembly (z-average hydrodynamic diameter is around 60 nm). As the protein load increases, those complexes assemble into multichain clusters, which retain their size (approximately 400 nm in diameter) in a relatively broad (30–90) protein-to-polymer molar ratio range (Figure 3A). ITC results clearly demonstrate that the ability of micron size complexes containing more than one hundred protein molecules per polymer chain to bind more lysozyme is rapidly decreasing (Figure 1C and Figure 3B). One might suggest that the increased binding of the positively charged protein decreases overall negative charge of the polymer, thus, lowering its protein binding propensity and facilitating the self-assembly to larger micron-sized complexes. It needs to be noted that ITC results also show some increase in the avidity of lysozyme to submicron complexes as the protein loading in them raises (region I in Figure 3B). A similar positive cooperativity effect was reported for interactions between low molecular weight PCPP and lysozyme in solutions with low ionic strength [44]. However, although still detectable, this effect for the high molecular weight polymer of the present study for formulations prepared at near physiological conditions appears to be not nearly that pronounced. 

Negative enthalpy and entropy changes determined by ITC using SSIS model (Figure A1) are typically associated with non-covalent binding of electrostatic and hydrophobic nature, rather than interactions driven by counter-ion release, which favor positive entropy change [3]. This is in line with the anisotropic model of interactions suggested previously [41,43]. According to the theory, complementarity between protein and its polyelectrolyte partner is mainly determined by charge anisotropy of protein surface and flexibility of polymer chain, rather than the release of counterions. The correlation between high protein-polyelectrolyte affinity and a large favorable binding enthalpy has been observed as a general trend, whereas the formation of low affinity complexes is typically driven only by entropy [41,43]. It has been also noted that highly flexible polymer backbone conforming to patches of charges on protein surfaces, to better describe protein-polymer complexation at higher salt concentrations than counterion release [3]. The above observations align well with high flexibility and high charge density of PCPP [10], which constitute a physico-chemical basis for its high degree of enthalpic ion-pairing with proteins.

A large numbers of protein molecules associated with single PCPP chain is consistent with previous findings for PCPP complexes with proteins, as detected by SEC and AF4 [12,14,15]. Notably, the dissociation constant calculated on the basis of ITC data is in the 10^−8^ M range, which is in line, but is somewhat lower than micromolar values detected by AF4 method for similar systems [12,22]. This may not be only associated with obvious differences in studied systems but can be potentially explained by non-specific interactions between the analyte and stationary phase, which are inherently present in methods, such as SEC or AF4, and can interfere with the analysis.

In the analysis of *w*NMR results, it is important to discuss potential mechanisms, which can be responsible for the sensitivity of water proton transverse relaxation rate, *R*_2_(^1^H_2_O) to solute organization. It is generally considered that the sensitivity of *w*NMR towards the presence of protein aggregates results from the proton exchange between water molecules and the exchangeable protons of a protein and its aggregates [40]. As protons of larger aggregates relax faster due to much longer rotational correlation time of the bulkier assembly, their proton exchange with water leads to a higher observed relaxation rate of water protons. Another potential factor of *w*NMR sensitivity pertinent to the present study, is the formation of water compartments inside the cavities of larger agglomerates. It is well established that the diffusive exchange of compartmentalized water differs from bulk water molecules and accelerates the experimentally observed water proton relaxation [37,39]. Third potential mechanism affecting *w*NMR observations is stipulated by the difference in magnetic susceptibilities of the solute and water (magnetic susceptibility contrast) [45]. In case of high magnetic susceptibility contrast, local magnetic field gradient is generated at the water-solute interface which accelerates spin de-coherence and results in the increase of *R*_2_(^1^H_2_O). However, in the present case, we do not expect the manifestation of this third mechanism, since the magnetic susceptibility contrast between water and proteins and organic polymers is typically extremely low [37]. One also can exclude the effects of viscosity changes on the observed *R*_2_(^1^H_2_O) values. Indeed, in the concentration range of lysozyme and PCPP used in this study (the highest being below 1.5 mg/mL), the changes in viscosity are hardly detectable (by viscometry or DLS autocorrelation function), therefore, will have no effect on water proton transverse relaxation.

It is important to note that *w*NMR approach has been already successfully applied to detect aggregation in solutions of therapeutically relevant monoclonal antibody [36,38], clustering of nanoparticles [37], and agglomeration in the aluminum-adjuvanted vaccine formulations under freeze–thaw stress [35]. 

The results of the present study demonstrate for the first time that applications of *w*NMR approach can be advantageously extended to exploring protein-polymer interactions. It is clear that the detected growth of *R*_2_(^1^H_2_O) in the process of PCPP titration with lysozyme cannot be associated with increasing concentration of protein (Figure 2A) but results solely from the binding of protein to polyelectrolyte, which occurs at near physiological conditions. Such interactions lead to the immobilization of smaller protein molecules on a larger polymer chain resulting in a longer rotational correlation time (*τ*_c_) of a bulky lysozyme-PCPP assembly, which, in turn, ensues faster relaxation of the lysozyme protons involved in the exchange with H_2_O. Some levelling-off of the *R*_2_(^1^H_2_O) values at a higher protein-to-polymer molar correlates well with the formation of micron-size aggregates and reduction in binding rates as detected by DLS and ITC, correspondingly (Figure 3A–C). It is also noteworthy to emphasize, that the use of *w*NMR allows detection of single chain complexes at a low protein loads, at which DLS—the commonly used analytical technique, fails to detect any changes (Figure 4).

The ‘reverse’ titration of lysozyme with PCPP, which leads to formulations with large excess of lysozyme (up to 350 molecules per PCPP chain) in the initial phase of the titration, can be expected to result in the immediate formation of micron-size aggregates. Under these conditions and given the reduced binding ability of such already protein-saturated aggregates, it can be anticipated that the results of both ITC and *w*NMR titrations, show minimal changes, if any, upon further addition of PCPP (Figure 2B,C and Figure A2). These observations can provide an important guidance for scientists interested in preparing high affinity complexes between proteins and polyelectrolytes and avoid largely heterogeneous formulations.

## 4. Materials and Methods

### 4.1. Materials

Lysozyme (Sigma-Aldrich, Saint Louis, MO, USA), phosphate buffered Saline (PBS), pH 7.4 (Thermo Fisher Scientific, Waltham, MA, USA) were used as received. High molecular weight poly[di(carboxylatophenoxy)phosphazene], PCPP (800,000 g/mol) was synthesized as described previously [46,47].

### 4.2. ITC Measurements

ITC experiments were performed using Nano ITC SV instrument (TA Instruments, Waters, New Castle, DE, USA) at 25 °C. In a typical experiment, 10 μL aliquots of titrant were injected from a 250 μL rotating syringe (400 rpm) into an isothermal chamber containing 900 μL of titrate with a 300 s delay between each injection. Each injection generated a heat burst curve (microcalories per second versus seconds), which was integrated using NanoAnalyze software, version 3.12.5 (TA Instruments, Waters, New Castle, DE, USA) to yield the heat associated with each injection. Data analysis was performed using the above software and SSIS fitting model to calculate reaction stoichiometry (n), binding constant (KD), enthalpy (∆H), and entropy (∆S).

### 4.3. wNMR Measurements

Experiments were performed at 25 °C using MQC+ (Oxford Instruments, plc., Abingdon, UK), at 23.8 MHz ^1^H resonance frequency, probe ID 26 mm, magnet and probe temperature 25 °C. MQC+ is equipped with built-in shimming system which provides the level of magnetic field homogeneity comparable or even better than high-field high-resolution NMR instruments. Additionally, Carr-Purcell-Meiboom-Gill (CPMG) pulse sequence with alternating phases of the 180°-pulses used in this study for transverse relaxation measurements is designed to compensate for any potential inhomogeneities of the external magnetic field, *B*_0_ [48,49]. Prior to measurements, samples in Nalgene^®^ cryogenic vials (0.5 mL sample volume) placed in the 18 mm borosilicate glass NMR tube which is inserted in 26 mm borosilicate glass NMR tube were equilibrated in the probe of MQC+ for 40 min. Such two tubes arrangement was used to place the sample in the Nalgene^®^ cryogenic vial in the center of NMR probe. Water proton transverse relaxation time *T*_2_(^1^H_2_O) was measured using CPMG pulse sequence with interpulse delay (*τ*) 500 µs and 18,000 echoes collected. The value of interpulse delay *τ* (500 µs) in the CPMG pulse sequence was selected based on the expected transverse relaxation time of water protons (~2.5 s) to exclude potential diffusion effects known to occur at long *τ* values. Relaxation measurements at variable *τ* (CPMG-dispersion or *τ*-dispersion) are typically used to explore the diffusion effects which we tried to avoid in this study. Two transients were accumulated with relaxation delay 12 s (total duration of one measurement ~1 min). 

Spin-echo signal intensity decay data were phased using RINMR ver. 7.0 software (Oxford Instruments, plc., Oxford, UK) and then were processed using single-exponential fitting function in WinFit ver. 2.4 software (Resonance Instruments, Ltd., Witney, UK) to extract water proton transverse relaxation time *T*_2_(^1^H_2_O)
*I(t) = I*_0_*× exp*[*−t/T*_2_(^1^H_2_O)]
where *I*(*t*) is the observed echo signal intensity over the echo decay time *t*, and *I*_0_ is a pre-exponential factor, echo signal intensity at *t* = 0. Relaxation time *T*_2_(^1^H_2_O) was converted to water proton transverse relaxation rate *R*_2_(^1^H_2_O) (*R*_2_(^1^H_2_O) = 1/*T*_2_(^1^H_2_O)). 

### 4.4. DLS Measurements

Dynamic light scattering (DLS) measurements were conducted using a Malvern Zetasizer Nano series instrument (Malvern Instruments Ltd., Worcestershire, UK) and analyzed using Malvern Zetasizer 7.10 software (Malvern Instruments Ltd., Worcestershire, UK). Samples were prepared in PBS (pH 7.4) and filtered using Millex 0.22 um filters prior to the analysis.

## 5. Conclusions

In vivo potency of PCPP—a clinical stage immunoadjuvant and vaccine delivery vehicle, is critically dependent on its ability to form and maintain supramolecular complexes with antigenic proteins under physiological conditions. Present study explored its interactions with a model antigen, lysozyme, in PBS at pH 7.4 using a combination of ITC, *w*NMR, and DLS methods. The supramolecular assembly under conditions of physiologically relevant salt concentrations is largely driven by the enthalpic factor, which is in agreement with previously suggested charge anisotropy theory and high flexibility of polyphosphazene backbone (small persistence length). The results also present *w*NMR as a new convenient analytical tool for exploring protein-polyelectrolyte interactions.

## Figures and Tables

**Figure 1 molecules-27-07424-f001:**
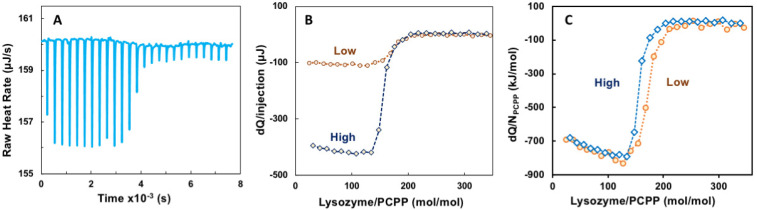
ITC data for the binding of lysozyme onto PCPP (direct titration). (**A**) raw data, (**B**) incremental heat per injection and (**C**) ITC isotherm for two different concentrations of reactants (0.125 (**A**–**C**) and 0.5 mg/mL (**B**,**C**) PCPP solutions were titrated with 2.5 and 10 mg/mL solutions of lysozyme, correspondingly. Concentrations are shown in Figures as “High” (blue, diamonds, dotted line) and “Low” (brown, spheres, pointed line); PBS, pH 7.4).

**Figure 2 molecules-27-07424-f002:**
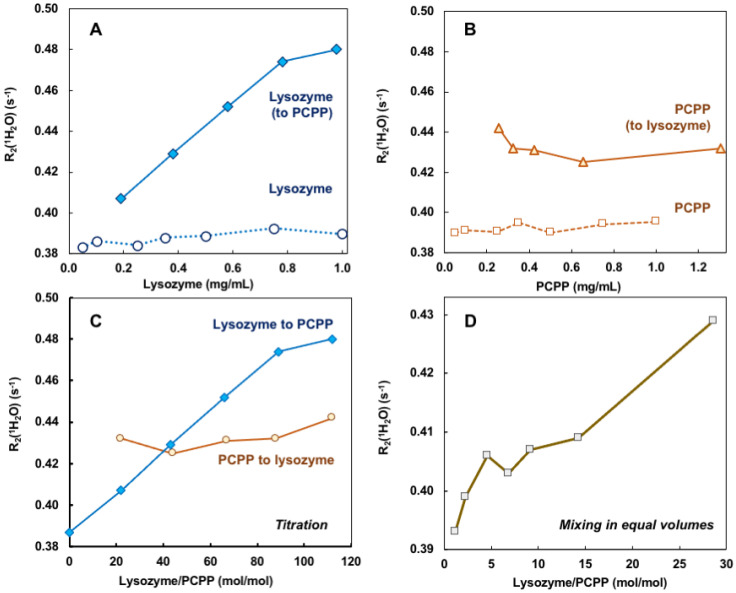
*w*NMR data for lysozyme–PCPP binding. (**A**,**B**) Concentration dependences of *R*_2_(^1^H_2_O) for lysozyme (**A**, spheres, dotted line), PCPP (**B**, squares, dashed line), and changes observed upon addition of lysozyme to PCPP (**A**, diamonds, solid line) and PCPP to Lysozyme (**B**, triangles, solid line) obtained by titration (10 mg/mL titrant was added to 0.5 mg/mL analyte, PBS, pH 7.4); (**C**) *w*NMR lysozyme-to-PCPP titration data (blue line, diamonds) and PCPP-to-lysozyme titration data plotted against lysozyme-to-PCPP molar ratio; (**D**) *w*NMR data for complexes obtained by mixing lysozyme and PCPP in equal volumes (PBS, pH 7.4).

**Figure 3 molecules-27-07424-f003:**
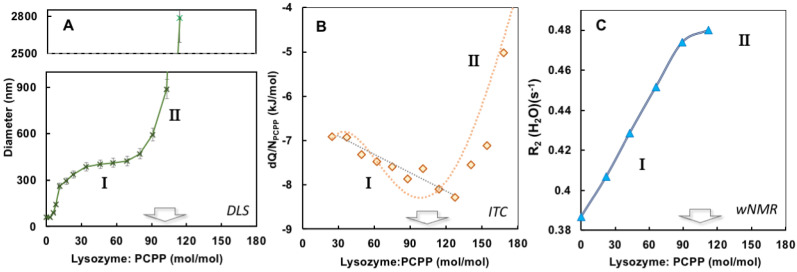
Comparison of DLS, ITC and *w*NMR titration results. (**A**) Dependence of z-average hydrodynamic diameters (**B**) incremental heat per polymer molar concentration, and (**C**) water proton transverse relaxation rate on the protein-to-polymer ratios (0.5 mg/mL PCPP was titrated with 10 mg/mL lysozyme, PBS, pH 7.4).

**Figure 4 molecules-27-07424-f004:**
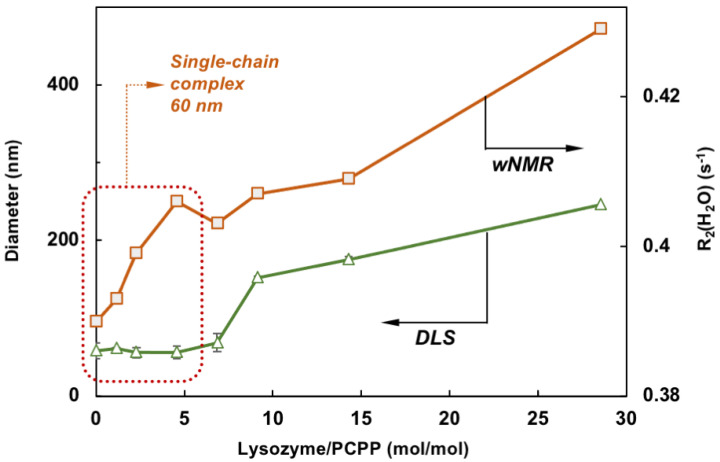
Comparison of *w*NMR and DLS data shows that contrary to DLS, *w*NMR is capable of detecting self-assembly processes at low lysozyme loadings (the zone for complexes formed with a single PCPP chain is designated by dotted line, z-average diameters are shown, mixing in equal volumes).

**Figure 5 molecules-27-07424-f005:**
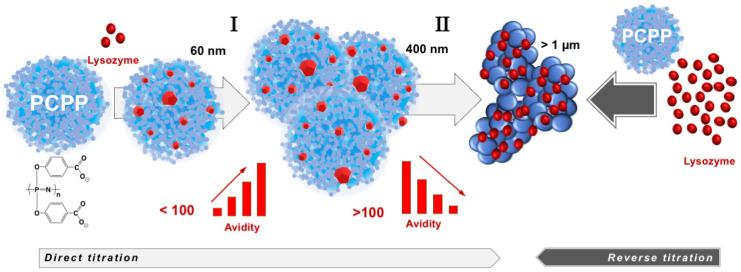
Lysozyme self-assembles onto PCPP with the formation of multimeric nano-scale, submicron size complexes and micron size aggregates with avidity varying as the self-assembly process proceeds (the drawings of complexes and aggregates are not to scale).

## Data Availability

Data available upon reasonable request.

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
