# Peer review of "Supramolecular Protein-Polyelectrolyte Assembly at Near Physiological Conditions—Water Proton NMR, ITC, and DLS Study"

_molecules, 2022, doi:10.3390/molecules27217424_

Round 1

Reviewer 1 Report

The authors studied the formation of supra molecular complexes of lysozyme and PCPP under the physiological environment along with the formation process. They utilized the ITC, wNMR and DLS to explore the self assembly process and the interaction between proteins and polyelectrolytes. They found that the assembly is due to the enthalpic factor and is in good agreement with the previous theories. This is a nice work and the logic flow is easy to understand. I recommend it to be accepted. 

Author Response

The authors studied the formation of supra molecular complexes of lysozyme and PCPP under the physiological environment along with the formation process. They utilized the ITC, wNMR and DLS to explore the self assembly process and the interaction between proteins and polyelectrolytes. They found that the assembly is due to the enthalpic factor and is in good agreement with the previous theories. This is a nice work and the logic flow is easy to understand. I recommend it to be accepted. 
Thank you!

Reviewer 2 Report

The Ms presents quite high potential interest for readers,

as it combines standard and new methods for the characterisation of

supramolecular assemblies.

A few suggestions:

- the authors should report the changes in viscosity in titrations

as well as potential changes in homogeneity

(could not find this in the Ms, and the magnetic homogeneity

impacts R2 )

- various CPMG delays should be used, instead of just one,

in order to rule out shimming and chemical exhange artefacts

(by the way, the articles of Carr & Purcell as well as Meiboom and Gill

should be cited)

style: the lack of definite articles in many placces in the Ms can confuse the

reader (concision should not be pursued at the expense of clarity)

Author Response

Authors are grateful to the reviewer for their valuable input, which has helped us to improve the quality of the manuscript.

Comments and Suggestions for Authors
The Ms presents quite high potential interest for readers, as it combines standard and new methods for the characterisation of supramolecular assemblies.
A few suggestions:
- the authors should report the changes in viscosity in titrations as well as potential changes in homogeneity (could not find this in the Ms, and the magnetic homogeneity impacts R2 )
Thank you. The following sentence was added to the Discussion section:
“One also can exclude the effects of viscosity changes on the observed R2(1H2O) values.  Indeed, in the concentration range of lysozyme and PCPP used in this study (the highest being below 1.5 mg/mL), the changes in viscosity are hardly detectable (by viscometry or DLS autocorrelation function), therefore, will have no effect on water proton transverse relaxation.”
- various CPMG delays should be used, instead of just one, in order to rule out shimming and chemical exhange artefacts (by the way, the articles of Carr & Purcell as well as Meiboom and Gill should be cited) style: the lack of definite articles in many placces in the Ms can confuse the reader (concision should not be pursued at the expense of clarity)
3 additional references were added, which also include those suggested by the reviewer: 
Ref. 45: Feng, Y.; Taraban, M. B.; Yu, Y. B., Water proton NMR—a sensitive probe for solute association. Chem. Commun. 2015, 51, (31), 6804-6807.
Ref. 48: Y. Carr and E. M. Purcell, Effects of diffusion on free precession in nuclear magnetic resonance experiments, Phys. Rev., 1954, 94, 630–638.
Ref. 49: S. Meiboom and D. Gill, Modified Spin‐Echo Method for Measuring Nuclear Relaxation Times, Rev. Sci. Instrum., 1958, 29, 688–691.
Section 4.3 in the Materials and Methods section was updated to include
“MQC+ is equipped with built-in shimming system which provides the level of magnetic field homogeneity comparable or even better than high-field high-resolution NMR instruments. Additionally, Carr-Purcell-Meiboom-Gill (CPMG) pulse sequence with alternating phases of the 180°-pulses used in this study for transverse relaxation measurements is designed to compensate for any potential inhomogeneities of the external magnetic field, B0 [48, 49].”
and
“The value of interpulse delay  (500 µs) in the CPMG pulse sequence was selected based on the expected transverse relaxation time of water protons (~2.5 s) to exclude potential diffusion effects known to occur at long  values.  Relaxation measurements at variable  (CPMG-dispersion or -dispersion) are typically used to explore the diffusion effects which we tried to avoid in this study.” 
The relevant addition was made in the Discussion section.

Reviewer 3 Report

1. The English should be polished.

2. These rencent researches of self assembly should be cited in introduction, such as Materials Today Advances, 2022, 14, 100231; Front. Bioeng. Biotechnol. 2022, 10, 903219.

Author Response

Authors are grateful to the reviewer for their valuable input, which has helped us to improve the quality of the manuscript.
Comments and Suggestions for Authors
1. The English should be polished.
Authors conducted extensive review of English and addressed all typos and errors.
2. These rencent researches of self assembly should be cited in introduction, such as Materials Today Advances, 2022, 14, 100231; Front. Bioeng. Biotechnol. 2022, 10, 903219.
New references are now added: Ref. 8 (V.S. Saji, Mater. Today Adv., 2022) and Ref. 9 (Wang, Front. Bioeng. Biotechnol.)